# Numerical Simulation Study on Flow Laws and Heat Transfer of Gas Hydrate in the Spiral Flow Pipeline with Long Twisted Band

**DOI:** 10.3390/e23040489

**Published:** 2021-04-20

**Authors:** Yongchao Rao, Lijun Li, Shuli Wang, Shuhua Zhao, Shidong Zhou

**Affiliations:** 1School of Petroleum Engineering, Changzhou University, Changzhou 213164, China; ryc@cczu.edu.cn (Y.R.); 19082003350@smail.cczu.edu.cn (L.L.); zsh@cczu.edu.cn (S.Z.); zsd@cczu.edu.cn (S.Z.); 2Key Laboratory of Oil and Gas Storage and Transportation Technology of Jiangsu Province, Changzhou 213164, China

**Keywords:** hydrate, spiral flow, long twisted band, heat transfer, numerical simulation

## Abstract

The natural gas hydrate plugging problems in the mixed pipeline are becoming more and more serious. The hydrate plugging has gradually become an important problem to ensure the safety of pipeline operation. The deposition and heat transfer characteristics of natural gas hydrate particles in the spiral flow pipeline have been studied. The DPM model (discrete phase model) was used to simulate the motion of solid particles, which was used to simulate the complex spiral flow characteristics of hydrate in the pipeline with a long twisted band. The deposition and heat transfer characteristics of gas hydrate particles in the spiral flow pipeline were studied. The velocity distribution, pressure drop distribution, heat transfer characteristics, and particle settling characteristics in the pipeline were investigated. The numerical results showed that compared with the straight flow without a long twisted band, two obvious eddies are formed in the flow field with a long twisted band, and the velocities are maximum at the center of the vortices. Along the direction of the pipeline, the two vortices move toward the pipe wall from near the twisted band, which can effectively carry the hydrate particles deposited on the wall. With the same Reynolds number, the twisted rate was greater, the spiral strength was weaker, the tangential velocity was smaller, and the pressure drop was smaller. Therefore, the pressure loss can be reduced as much as possible with effect of the spiral flow. In a straight light flow, the Nusselt number is in a parabolic shape with the opening downwards. At the center of the pipe, the Nusselt number gradually decreased toward the pipe wall at the maximum, and at the near wall, the attenuation gradient of the Nu number was large. For spiral flow, the curve presented by the Nusselt number was a trough at the center of the pipe and a peak at 1/2 of the pipe diameter. With the reduction of twist rate, the Nusselt number becomes larger. Therefore, the spiral flow can make the temperature distribution more even and prevent the large temperature difference, resulting in the mass formation of hydrate particles in the pipeline wall. Spiral flow has a good carrying effect. Under the same condition, the spiral flow carried hydrate particles at a distance about 3–4 times farther than that of the straight flow.

## 1. Introduction

The natural gas hydrate plugging problems in the mixed pipeline are becoming more and more serious. The hydrate creates natural gas pipeline blockage, and the partial pressure of the pipeline damages pipeline equipment. The hydrate plugging gradually has become an important problem to ensure the safety of pipeline operation [1,2]. The traditional methods to prevent hydrates from blocking pipelines are to change the formation conditions of hydrates by heating and lowering the pressure or to inhibit the formation of hydrates by injecting thermodynamic inhibitors. These methods not only cost a lot but also cause pollution to the environment. The current research is not focused on hydrate inhibition, but the generation of hydrate is OK; however, we must ensure the safe flow of hydrate.

There are many kinds of spiral devices, such as a spiral twisted band, impeller, and spiral guide bar. According to the different use conditions and purposes, different spiral devices are needed. There are three methods for the conventional generation of spiral flow: tangential inlet flow, installation of flow guide and spiral pipe. Wang et al. [3,4] made a comparative analysis of various spiral flow rotating devices and carried out experimental studies. Team [5] used PHOENICS computational fluid software to numerically simulate the formation and attenuation of spiral flow in a pipeline with a velocity angle of 5°–70°. It is found that the spiral flow has a suitable tangential and axial velocity when the velocity angle changes between 5° and 30°. The spiral flow is also conducive to the removal of deposited impurities and water in the pipeline. Experimental results show that gas-liquid spiral flow fails to endure all the time downstream of the swirler and finally transforms to non-swirling flow [6]. Zhang et al. [7] studied the attenuation characteristics of the spiral flow of the flat axis round tube generated by the local generator through experiments, and they concluded that the tangential velocity and the strength of the spiral flow decreased with the increase of the distance. In conclusion, spiral flow not only has strong carrying capacity but also has important application value for safe transportation of NGH (natural gas hydrate). The research group also used the RNG (re-normalization group) k-ε model to numerically simulate the flow characteristics of swirl flow rotated by the vane in the horizontal pipe [8]. The Reynolds number had a great influence on spiral intensity. The spiral intensity would increase and the spiral intensity attenuation rate would decrease when the Re number increased. Moreover, the spiral flow can enhance the heat transfer between phases, which is a controllable technique for the formation and prevention of hydrate in the pipeline. The average thickness of liquid film in the swirling annular flow increases compared with the annular flow [9,10,11]; this phenomenon provides benefits for increasing critical heat flux to enhance heat transfer efficiency [12,13].

Researching the safe flow of natural gas hydrate, Song et al. [14] introduced a population equilibrium model based on the aggregation dynamics of hydrate particles to simulate the effects of fluid flow rate and hydrate volume fraction on the safe flow characteristics of hydrate slurry. Then, the population balance model based on solid hydrate grain accumulation dynamics [15], the collision frequency, aggregation efficiency, fragmentation frequency, and sub-particles size distribution function after crushing of the hydrate solid particles were considered in the flow process. The simulation results were compared with the calculated results of the hydrate particle growth model. In addition, Song et al. [16,17] simulated the deposition rule of gas hydrate particles in the pipeline by referencing the condensation method, and he established the kinetic model of hydrate deposition and hydrate decomposition. Based on the theory of hydrate particle deposition, the structural model of hydrate deposition was established, the main mechanical parameters were determined, and the finite element numerical simulation method was used to calculate the hydrate deposition on the pipe wall. Through the establishment of a three-dimensional model of circular pipe spiral flow, the RNG k-ε turbulence model and DPM model were used to simulate the three-dimensional transient of gas–solid two-phase spiral flow and heat transfer in natural gas pipeline [18,19], and the velocity field, temperature field, distribution law of hydrate particle volume fraction, and heat transfer law of different cross-sections in a natural gas pipeline were studied. Experimental research on characteristics of the gas–solid two-phase flow spun up by the twisted band in pipelines was conducted [20], and the effect of gas phase velocity and parameters of twisted band on the particle carrying laws, collision characteristics, carrying distance, particle velocity distribution, and particle concentration distribution were investigated, and flow characteristics of gas–solid two phase non-spiral flow and gas–solid two phase spiral flow were compared and analyzed. Cai et al. [21] used the DPM model and RSM (Reynolds Stress Model) model to simulate the spiral transport of gas hydrate particles with a twisted band. The temperature field, velocity field, turbulence intensity, and deposition law of hydrate particles under different twisted rates and flow rates were examined. Chang et al. [22] carried out a three-dimensional transient numerical simulation of gas–solid two-phase spiral flow in natural gas pipelines, and they studied the volume concentration variation characteristics of hydrate particles in natural gas pipelines and the deposition characteristics of hydrate particles. Considering the formation process of hydrate, Liu et al. [23] established the equation of mass, momentum, and energy balance. The iterative method and finite difference method were used to solve the model results, and the sensitivity analysis of the important parameters of the model is carried out. Brown et al. [24] used a micromechanical force meter to explore the interaction between the wax deposited on the surface and dissolved in the bulk phase and the hydrate chemically treated to prevent agglomeration. It was found that wax can significantly change the cohesion and adhesion of hydrate particles, but the effect of anti-caking agent may vary. Nicholas et al. [25] explored the deposition of saturated natural gas condensate containing water on the pipeline wall by using a single-circulation pipeline, and he found that the deposition of hydrate on the wall could cause the pipeline pressure drop to rise slowly. At the same time, the wall sediments can insulate the fluid inside the tube, and it made the wall sediments migrate downstream. Lorenzo et al. [26] studied the growth and deposition of hydrate on the wall under the condition of annular flow with a single circulation pipeline, and he found that the membrane growth model under the condition of low subcooling could well predict the pressure drop of the loop after hydrate formation. However, other particle behavior characteristics, such as the shedding of the hydrate layer, should be considered under the condition of high supercooling. A numerical investigation is done on the effect of employing the new combined vortex generators, the twisted tape turbulator, and Al_2_O_3_-H_2_O nanofluid as the involved base fluid. The results show that the pitch ratio has a predominant effect on the Nusselt number and the friction factor, which causes an efficiency increase of up to five times more than the original one [27]. The work focuses on the design of experiments based on Response Surface Methodology to determine the optimum values for maximizing the heat transfer and minimizing the friction factor in a DPHE (Double Pipe Heat exchangers) inserted with varying cross-section cut twisted tapes. The combined effect of radius and angle of cut on HTC and friction factor is explored, and optimal parameters of tape inserts and flow are obtained [28].

In conclusion, academics have carried out the numerical simulation and experimental research of the combination of pipeline flow law of gas hydrate, but the flow law of hydrate particles with long twisted band is little. At the same time, the heat transfer of the spiral flow system of the law on hydrate formation and the influence of the particle movement are also the important interference factors. Therefore, the DPM model is used. The movement of solid particles is used to numerically simulate the complex spiral flow characteristics of hydrates in the spiral flow pipeline with a long twisted band. The deposition and heat transfer characteristics of natural gas hydrate particles in the spiral flow pipeline would have been studied. The velocity distribution, pressure drop distribution, heat transfer characteristics, and particle settling characteristics would have been investigated using the Computational Fluid Dynamics (CFD) technology.

## 2. Numerical Simulation Method

### 2.1. Physical Model

The formation of hydrate requires a series of complex processes and does not exist directly in the pipeline. In this paper, a series of reaction processes of hydrate formation are ignored in the numerical simulation, and it is assumed that hydrate particles have been generated directly at the entrance of the pipeline. In the numerical simulation, the shape of hydrate particles is set as a positive circle and the particle size is also set as the same, while the influence of pipe wall thickness on the formation of hydrate is directly ignored in the calculation. The calculation model established by CAD (Computer-Aided Design) software is shown in Figure 1. The pipe diameter D = 0.024 m and the length L = 2.5 m. At the same time, a group of hydrate particles with partial rotation in the short twisted band is set as a comparison, and the length of the short twisted band is 0.5 m.

#### 2.1.1. Geometric Model

A twisted band with different twisted rates is set at the pipe inlet as the swirler. The physical model of the twisted band is shown in Figure 2, the twisted rates of the spiral twisted band are 6.2, 7.4, and 8.8, respectively. The schematic diagram of the twisted rate of the twisted band is shown in Figure 3. The twisted rate is the ratio between the length H of the twisted band and the width D of the twisted band after a rotation.

#### 2.1.2. Boundary Conditions

Before numerical simulation, the boundary conditions of the geometric model should be set reasonably. In the inlet end of the pipe, the velocity inlet is set as the boundary condition. The Reynolds Numbers is 5000, 10,000, 15,000, and 20,000, respectively. In the outlet, outflow exports are the boundary conditions, no slip wall is set to a fixed, and the problem is simplified to three-dimensional, constant physical gas–solid flow. The inlet temperature is 280 K, and the wall temperature Tw is 277 K. The model is calculated using rectangular coordinate system. The origin is at the center of the entry interface, the z-axis is the flow direction, the gravity direction is along the y-axis, and the gravity acceleration is 9.81 m/s^2^. The fluid medium is natural gas and hydrate particles, flowing from the left end of the pipeline to the right.

#### 2.1.3. Initial Conditions

The diameter of the gas pipeline (D) is 0.024 m, and the length (L) is 2.5 m. In the simulation, it is considered that the physical properties of the hydrate in the spiral flow pipeline are constant, the gas phase is methane, and the solid phase is NGH particles. The basic physical parameters are measured at room temperature. Firstly, the natural gas density is 0.77 kg/m^3^, the kinematic viscosity of natural gas is 11.03 × 10^−6^ m^2^/s, and the hydrate particle density is 915 kg/m^3^. The initial volume concentration of hydrate is 1%, 2%, 4%, 6%, and 8% in DPM, and the movement of particles at different concentrations is simulated. The temperature of the fluid and hydrate particles in the natural gas pipeline is the same, and there is no temperature difference. Moreover, the hydrate particles are homogeneous spheres with the same particle size in the simulation. Data parameters are selected according to the research content, as shown in Table 1.

### 2.2. Meshing

The calculation grid of the pipe section with the long twisted band is shown in Figure 4. The entire geometric model is meshed. As a result of the twisted band, the unstructured grid is applied, and the dense processing of the pipe wall can better perform the turbulence calculation. The most suitable grid computing is selected through the grid independence test, and the number of grid cells is 4 million.

### 2.3. Mathematical Model

The DPM model is used in the simulation. The discrete phase model is a multi-component flow model under the Euler-Lagrangian method. In the simulation, the solid particles are treated as discrete phase, while the fluid is treated as continuous phase. When the fluid continuous phase is stabilized in the calculation, the particle discrete phase is introduced. In the discrete phase model, the particle phase is regarded as the discrete phase, while the fluid is only regarded as the continuous phase. The particle volume fraction is the mass transfer from phase q to phase p.

The energy equation:(1)$$\frac{\partial \left(\rho T\right)}{\partial t}+\frac{\partial \left(\rho uT\right)}{\partial x}+\frac{\partial \left(\rho vT\right)}{\partial y}+\frac{\partial \left(\rho wT\right)}{\partial z}=\frac{\partial }{\partial x}\left(\frac{\lambda }{c_{p}}\frac{\partial T}{\partial x}\right)+\frac{\partial }{\partial y}\left(\frac{\lambda }{c_{p}}\frac{\partial T}{\partial y}\right)+\frac{\partial }{\partial z}\left(\frac{\lambda }{c_{p}}\frac{\partial T}{\partial z}\right)$$
where, *ρ*, $c_{p}$, *T*, and λ are the gas density, specific heat capacity at constant pressure, temperature, and thermal conductivity ($c_{p}$ = 2.205 kJ/kg⋅k, λ = 0.14 W/m⋅k), *u*, *v*, and *w* are all velocities, and *t* is time.

#### 2.3.1. Governing Equations

The equation of continuity,
(2)$$\frac{\partial \rho }{\partial t}+\frac{\partial }{\partial x_{i}}\left(\rho u_{i}\right)=0$$
The momentum equation,
(3)$$\frac{\partial }{\partial t}\left(\rho u_{i}\right)+\frac{\partial }{\partial x_{j}}\left(\rho u_{i}u_{j}\right)+\frac{\partial p}{\partial x_{i}}-\frac{\partial \tau_{ij}}{\partial x_{j}}-\frac{\partial \tau_{ij}{}^{-1}}{\partial x_{j}}=0$$
where, *ρ*, *u*, and *p* are gas density, velocity, and static pressure, respectively; *τ_ij_* is the viscous stress tensor; and *t* is the time. Volume fraction equation:(4)$$\frac{1}{\rho_{q}}\left[\frac{\partial }{\partial t}(a_{q}\rho_{q})+\nabla \cdot (a_{q}\rho_{q}{\bar{v}}_{q})=S_{a_{q}}+\sum_{p=1}^{n}\left({\dot{m}}_{pq}-{\dot{m}}_{qp}\right)\right]$$
where $a_{q}$ is the volume fraction of phase q, $S_{a_{q}}$ is the source term, ${\dot{m}}_{pq}$ is the mass transfer from phase p to phase q, ${\dot{m}}_{qp}$ is required to be relatively small, and the solid particle concentration of the discrete phase is generally below 10%.

#### 2.3.2. Turbulent Motion Equation

The standard k-ε model is a semi-empirical formula. The equation is:(5)$$\frac{\partial }{\partial t}\left(\rho k\right)+\frac{\partial }{\partial x_{i}}\left(\rho ku_{i}\right)=\frac{\partial }{\partial x_{j}}\left[\left(\mu +\frac{\mu_{t}}{\sigma_{k}}\right)\frac{\partial k}{\partial x_{j}}\right]+G_{k}-Y_{k}+S_{k}$$

(6)$$\frac{\partial }{\partial t}\left(\rho \varepsilon \right)+\frac{\partial }{\partial x_{i}}\left(\rho \varepsilon u_{i}\right)=\frac{\partial }{\partial x_{j}}\left[\left(\mu +\frac{\mu_{t}}{\sigma_{k}}\right)\frac{\partial \varepsilon }{\partial x_{j}}\right]C_{1\varepsilon }\frac{\varepsilon }{k}\left(G_{k}+C_{3\varepsilon }G_{b}\right)-C_{2\varepsilon }\rho \frac{\varepsilon^{2}}{k}+S_{\varepsilon }$$
where:(7)$$\mu_{t}=\rho C_{\mu }\frac{k^{2}}{\varepsilon }$$
among which $C_{1\varepsilon }$ = 1.44, $C_{2\varepsilon }$ = 1.92, $C_{\mu }$ = 0.09, $\sigma_{k}$ = 1.0, $\sigma_{\varepsilon }$= 1.3.

#### 2.3.3. Discrete Phase Model

The calculation of the discrete phase model mainly consists of a continuous phase and dispersed phase. The DPM model can obtain the particle motion equation by calculating the forces on particles, and the particle motion trajectory can be obtained by integrating the particle motion equation of the DPM model with time. The motion equation of the trajectory of solid particles in the z-direction is:(8)$$\frac{d_{u_{p}}}{d_{t}}=F_{D}(u-u_{p})+\frac{g_{z}(\rho_{p}-\rho )}{\rho_{p}}+F_{z}$$
where *u* is the velocity of the fluid phase, m/s; $u_{p}$ is the particle velocity, m/s; $g_{z}$ is the component of the gravitational acceleration in the z direction, m/s^2^; ρ is the fluid density, (kg/m^3^); $\rho_{p}$ is the particle density (kg/m^3^). The particles are mainly subjected to other forces $F_{z}$, including additional mass forces, Brownian forces, and Staffman lift forces et al. The Staffman lift force is caused by fluid acceleration around the particles. The additional mass forces are mainly applied when the density of the fluid phase is greater than that of the particles. The particle density of hydrate in the pipeline is greater than that of the airflow, so it is not necessary to consider the additional mass forces.

Here, $F_{D}$ is defined as:(9)FD=18μρpdp2CDRe24

Re is defined as the relative Reynolds number as follows:(10)Re=ρdp|up−u|μ
where $d_{p}$ is particle diameter, m.

The drag coefficient $C_{D}$ can be expressed as follows:(11)CD=a1+a2Re+a3Re2
$a_{1}$, $a_{2}$ and $a_{3}$ to constant values are as follows: 5000 < Re < 10,000 are respectively 0.46, −490.546, 578,700; when Re > 10,000, it is 0.5191, −16,625.5, and 5,416,700 respectively.

### 2.4. Calculation Method

CFD software is usually used to solve practical problems in numerical simulation. The software is generally composed of three parts: pretreatment, calculation and data solving, and post-processing. Pre-processing is to build a physical model and then conduct mesh division. The computing part in the middle is particularly important. Many parameters and algorithm selection need to be set before re-operation, among which the selection of algorithm can improve the accuracy of simulation and computing efficiency. The calculation can be roughly divided into the following steps: physical model selection, parameter setting (grid unit selection, material definition, phase selection, boundary condition setting, reference value setting, etc.), algorithm selection and control factor setting, initialization, monitor setting, and time step setting.

The discrete phase model is selected in the calculation, and the pressure base and implicit solver are used to simulate the transient state of the gas–solid three-dimensional helical flow in the gas hydrate pipeline. In the simulated calculation for discrete phase, the SIMPLEC algorithm is adopted to the coupling of pressure and velocity with the RNG k-Ɛ turbulence model, and a multidimensional linear reconstruction method is used to reconstruct the second-order surface pressure scheme, and the particle motion model is the DPM model. In the DPM model, it is also necessary to set some parameters at the inlet of the pipeline, such as the injection particle speed, mass flow rate and injection particle number, etc. During the iterative calculation in the simulation, the relaxation factor numerical range also needs to be set down in order to calculate the effect of better: $\varepsilon_{p}$ = 0.3–0.7, $\varepsilon_{m}$ = 0.5–0.7, $\varepsilon_{k}$ = $\varepsilon_{\varepsilon }$ = 0.4–0.6. When the absolute value of the residual is below 1 × 10^−6^, the condition of convergence can be achieved.

### 2.5. Numerical Examples Validate

#### 2.5.1. Grid Independence Test

Mesh generation is an important step in the establishment of a finite element model. In this process, many problems need to be considered, and the workload is very heavy. The unit length, quantity, and density of the grid have a direct influence on the precision and time of calculation.

In order to meet the requirements of computational accuracy and ensure computational efficiency, 8 mm, 4 mm, and 2 mm mesh sizes are selected for grid independence verification. The simulation verification conditions are as follows: pipe diameter D = 24 mm, the gas Reynolds number Re is 20,000, the twist rate of twisted band is 6.2, and the initial concentration of particles is 2%. As shown in Figure 5, the velocity distribution on the cross-section at Z = 5D of the pipeline is selected for comparison. Under these three mesh size conditions, the velocity distribution curves obtained by the simulation are generally similar. However, compared with 2 mm (6,541,200 cells) and 4 mm (4,209,480 cells), the mesh size of 8 mm (2,865,700 cells) varies greatly, and there are fewer meshes near the wall, so the calculation accuracy is not enough. On the other hand, the grid size of 2 mm (6,541,200 cells) and 4 mm (4,209,480 cells) is basically the same, and the accuracy is not improved much near the wall. In order to improve the computing efficiency, the grid size of 4 mm (4,209,480 cells) is finally selected as the computational grid.

#### 2.5.2. Experimental Verification of Gas–Solid Two-Phase Spiral Flow and Heat Transfer

The numerical simulation results of gas–solid two-phase spiral flow are compared with the experimental results to verify the feasibility. The results of the validation are shown in Figure 6 and Figure 7, and the pressure drop of the fluid (ΔP) and Nusselt number (Nu) with the Reynolds number (Re) changes in gas–solid two-phase flow. The pipe length is 1.2 m, the pipe diameter is 24 mm, and the particle size is 0.02 mm. The concentration of solid particles used in the simulation is the rate of solid volume to gas volume. From the simulation and experimental verification results, the error is very small, so the gas–solid two-phase flow in the gas hydrate pipeline can be calculated by the numerical simulation method.

The Reynolds number is calculated as follows:(12)Re=ρvdμ
where d is the diameter of the pipe, v is the average velocity of particle flow, ρ and μ are the density and dynamic viscosity of the gas, respectively.

## 3. Results

The deposition and heat transfer of particles in spiral pipe flow are studied. The velocity distribution, pressure drop distribution, particle deposition characteristics, and heat transfer characteristics in the natural gas pipeline are mainly studied, and the effects of long twisted band and no twisted band as well as different Reynolds numbers are compared and analyzed.

### 3.1. Velocity Distribution Law

The velocity distribution cloud map at each section under the condition of Re = 15,000 and different twisted rates is shown in Figure 8. It can be seen from the figure that the velocity is maximum in the central area of the pipe in the light pipe without a twisted band, and then velocity decreases uniformly toward the pipe wall, and the velocity change at each section is not obvious. However, two distinct eddies are formed on both sides of the twisted band with the maximum velocity at the center of the vortex. Along the direction of the pipe, the two vortices move from the near twisted band toward the wall. Compared with the short twisted band, the vortex always exists, and the reason is that the long twisted band is full of the whole pipe. When the short twisted band is used, the velocity tends to decrease as the particle transportation distance increases. When the twisted rate is the lower, the velocity of the two vortex centers is greater.

The velocity and vector graphs at different sections are shown in Figure 9. After the velocity distribution is stabilized, the velocity center can be observed at the far end of the pressure surface of the pipeline, and there is an obvious wake. It can be seen from the vector line that the tangential velocity first increases and then stabilizes. The tangential velocity hardly attenuates in the process of hydrate transportation, and the tangential velocity value near the wall is the largest. It is the reason that the tangential velocity is larger, that the hydrate particles tangential force with the wall is larger; the hydrate particle is not easy to deposit, and the hydrate particle is not prone to aggregation and deposition. The constant tangential velocity can further expand the safe flow boundary of hydrate.

### 3.2. Distribution Law of Spiral Flow Pressure Drop

#### 3.2.1. Influence of Reynolds Number on Pressure Drop

The relationship between pressure drop and Reynolds number is shown in Figure 10, when the twisted rate is 6.2. It can be seen from the figure that the pressure drop increases with the increase of Reynolds number at the same section of the pipe, and when the Reynolds number is higher, the pressure drop increases more. Along the direction of the pipe, the pressure decreases more obviously with the increase of the particle transportation distance, and the slope of the curve in the figure is larger, which is similar to a parabola.

#### 3.2.2. Influence of Twisted Rate on Pressure Drop

The relationship between Reynolds number and pressure drop under different twisted rates in horizontal pipeline is shown in Figure 11. It can be seen from the figure that the pressure drop is affected by the twisted rate. The twisted rate is smaller, and the pressure drop is greater. The twisted band has an influence on the pressure drop significantly. The reasons are as follows: the twisted band produces resistance to the fluid compared with the straight light pipe flow, and the spiral flow generated by the twisted band disturbs the boundary layer of the pipe and causes turbulent flow. As a result, the pressure drop is higher than the pipe without a twisted band.

In the case of constant particle Reynolds number, the twisted rate is greater, the spiral strength is weaker, the tangential velocity is smaller, the axial velocity loss is smaller, and the increase of pressure drop is smaller.

### 3.3. Analysis of Heat Transfer Law

#### 3.3.1. Axial Temperature Distribution Law

The temperature distribution cloud map at each section of the pipe with Re = 20,000 and different twisted rates is shown in Figure 12. From the figure, the temperature distribution in the pipe without twisted bands is relatively uniform and layered at the beginning. Along the direction of the pipe, the temperature from the center of the pipe to the wall decreases gradually due to the effect of heat transfer, and the temperature in the center of the pipe is the highest. After adding the twisted band, two vortices will be formed on both sides of the twisted band, the temperature at the two vortex centers is the highest, and the temperature gradually decreases from the center to the pipe wall. With the increase of the twisted rate, the temperature in the vortex center is relatively smaller. The twisted rate is smaller, the spiral flow generated is stronger, and the boundary layer disturbance is stronger. The temperature drops slowly compared with the short twisted band.

#### 3.3.2. Axial Distribution Law of Nusselt Number

The change curve of the average Nusselt number Nu with Re in the long twisted pipe is shown in Figure 13. The Nu changes little without a twisted band, and it approximates a horizontal line when Re is very small. A long twisted band is placed in the pipeline, and when the fluid flows through the internal twisted band, it will rotate, which enhances the tangential velocity of the fluid and continuously scours the boundary layer of the pipeline. Meanwhile, the disturbance of the boundary layer is also strengthened, and the turbulence intensity of the fluid near the wall is improved. The fluid tangential motion makes the boundary layer and the main body mix and strengthen, which effectively enhances the convective heat transfer of the fluid. From the figure, the twisted rate is smaller, and the Nu is larger. The reason is that the twisted rate is lower, and the spiral flow is longer. In the case of the same mass flow rate, the path of the fluid flow is longer, the spiral flow formation and velocity is larger, and the turbulence intensity of the fluid is greater. Therefore, the heat transfer is enhanced. The Nu distribution curve of all sections under the condition of different twisted rates when Re = 20,000 is shown in Figure 14. From the figure, Nu of the straight light pipe flow is almost unchanged along the pipe. When the twisted band is placed in the pipe, the Nusselt number first strengthens and then decreases after about 20D. Since the twisted band generates spiral flow at the beginning, which enhances the heat transfer of the pipe, Nu starts to become smaller with the heat exchange between the fluid and the pipe wall. The long twisted band fills the whole pipe compared with the short twisted band, so the spiral flow is maintained all the time, and the attenuation of the Nusselt number is relatively small.

#### 3.3.3. Radial Distribution Law of Nusselt Number

The Nu distribution curve on the section Z = 5D when Re = 20,000 is shown in Figure 15. In a straight light pipe flow without a twisted band, the Nusselt number is in a parabolic shape with the opening downwards. At the center of the pipe, the Nusselt number gradually decreases toward the pipe wall at the maximum, and the attenuation gradient of Nu is large. The reason is that there is no spiral disturbance in the straight light pipe flow, the temperature at the center of the pipe is the highest, and the heat exchange temperature of the pipe wall decreases gradually. In the pipe with a twisted band, the curve of Nusselt number appeared as a trough in the center of the pipe and a peak at half of the pipe diameter. The two eddies form on both sides of the twisted band, which enhances the maximum heat transfer temperature of the fluid, and the eddies gradually attenuate toward the tube wall and the twisted band. With the reduction of the twisted rate, the Nusselt number becomes larger.

### 3.4. Particle Deposition Law

The concentration distribution of hydrate particles at different positions is shown in Figure 16. It can be seen that the concentration distribution of hydrate particles in the initial section is relatively uniform with the long twisted band. With the increase of distance, the concentration of hydrate particles presents an obvious symmetrical center distribution, and the hydrate particles are flung to the near pressure surface of the twisted band due to centrifugal force. At the same time, it is located in the center of the spiral vortex, and the pressure is low, forcing the surrounding particles to concentrate to the low pressure area. The twisted rate is smaller, and the concentration area is more obvious. In the absence of twisted bands, particles begin to deposit significantly at the position of 0.7 m and reach the maximum particle concentration at the position of 1.5 m, and it indicates that this point is a possible location for blockage. When the spiral flow is not completely attenuated, the concentration of particles in each section is almost the same, and the concentration of particles in the pipe wall reaches the maximum and is evenly distributed. However, there are some obvious deposits at the 1.6 m position after the spiral flow attenuation. The transport distance has been increased by more than two times compared with the condition without swirler. However, the concentration distribution in all sections remains almost the same during the whole spinning process, and it indicates that no obvious deposition of particles occurs.

The distribution curve of the volume fraction of particles along the pipe when Y = 6.2 and α_0_ = 1% is shown in Figure 17. The figure shows that the volume fraction of particle is how to deposition in the long twisted band under three different Reynolds Numbers of Re = 3000, 5000, and 10,000. When Re = 3000, the curve has a peak value at Z = 56D approximately. The peak value represents the gradual accumulation of particles here, and the point is considered as the deposition point of particles. With the increase of Reynolds number, when the curve tends to the horizontal line with no peak value at Re = 5000 and 10,000, it can be considered that the particles are not deposited in the pipe. The curve of Re = 5000 is slightly higher than that of Re = 10,000, which indicates that the Reynolds number is larger and the deposition is more difficult. The curve of Re = 10,000 rises slightly because of the centrifugal motion of the particles in the long twisted band, some of the particles will stick to the wall of the tube, and the tangential force of the spiral flow is not enough to make it fly away. When Y = 6.2, the initial concentration α_0_ = 1%, and the critical deposition Reynolds number of particles is within the range from 3000 to 5000. The volume fraction distribution curve of particles along the pipeline when Y = 6.2 and α_0_ = 8% is shown in Figure 18. The particles are deposited when Re = 4000, and it indicates that the critical deposition Reynolds number of the particles is from 4000 to 5000 when Y = 6.2 and initial concentration α_0_ = 8%. The particle concentration is higher, and the deposition is easier. Comparing the two pictures, it can be concluded that the particle concentration is greater, the particle deposition is closer to the pipe, and there are more deposited particles.

The distribution curve of volume fraction of different twisted band particles along the pipe when Re = 4000 and α_0_ = 8% is shown in Figure 19. The short twisted band and long twisted band as well as a flat light tube without a twisted band are compared respectively, and the curve in the figure shows an increasing trend. In the straight light tube flow without a twisted band, the particles are deposited at Z = 12D, where the volume fraction of the particles is the largest. In the short twisted band with Y = 6.2, particles are deposited at Z = 28D, and the peak value of the curve is smaller than that without a twisted band. In the long twisted band with Y = 6.2, the deposition location of particles is approximately at Z = 42D, and the deposited particles in the pipeline are the least. It can be seen that there is the best carrying effect with a long twisted band whose carrying distance is about 3–4 times that of the non-twisted band. This result provides theoretical guidance for the safe transportation of hydrate. In Figure 19, Y_L_ is the twist rate of the long twisted band, and Y_S_ is the twist rate of the short twisted band.

## 4. Conclusions

(1)In the plain direct current of a light pipe without a twisted band, the velocity is maximum in the central area of the pipe and then decreases uniformly toward the pipe wall, and the velocity change at each section is not obvious. However, two distinct eddies are formed on both sides of the twisted band with the maximum velocity at the center of the vortex. Along the direction of the pipe, the two vortices move from the near twisted band toward the wall, and it can effectively carry the hydrate particles deposited on the pipe wall.(2)At the same section of the pipe, the pressure drop of the pipe increases with the increase of Reynolds number; when the Reynolds number is higher, there are greater increases of pressure drop. Along the direction of the pipeline, the pressure drop becomes more obvious with the increase of particle transport distance, and the pressure drop curve is similar to a parabola. In the case of constant particle Reynolds number, the twisted rate is greater, the spiral flow strength is weaker, the tangential velocity is smaller, the axial velocity loss is smaller, and the pressure drop is smaller. Therefore, the pressure loss can be reduced as much as possible while the effect of spiral flow can be guaranteed.(3)In a straight light pipe flow without a twisted band, the Nusselt number is in a parabolic shape with the opening downwards. At the center of the pipe, the Nusselt number gradually decreases toward the pipe wall at the maximum, and the attenuation gradient of Nu is large at the near wall. The curve of Nusselt number appears as a trough in the center of the pipe and a peak at half of the pipe diameter with twisted tape. With the reduction of the twisted rate, the Nusselt number becomes larger. Spiral flow can make the temperature distribution of the flow field in the pipeline more even and prevent the large number of formation of hydrate particles in the pipeline wall due to the large temperature difference.(4)The Reynolds number is larger, and the particles are less likely to deposit. The particle concentration is larger, and the particle deposition is closer to the pipe. When the twisted rate is 6.2, the initial concentration is between 1% and 8%, and the critical deposition Reynolds number is between 3000 and 5000. The spiral flow carries the hydrate particles a distance that is 3–4 times greater than that without a twisted band.

## Figures and Tables

**Figure 1 entropy-23-00489-f001:**
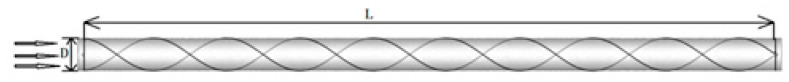
The physical model.

**Figure 2 entropy-23-00489-f002:**
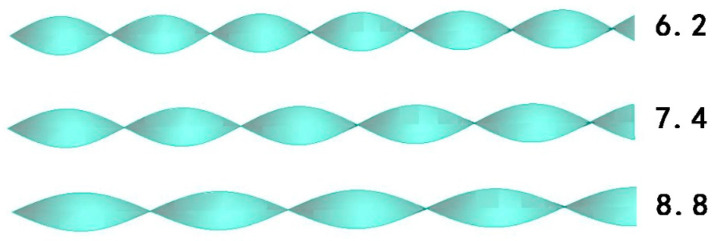
Twisted band model.

**Figure 3 entropy-23-00489-f003:**
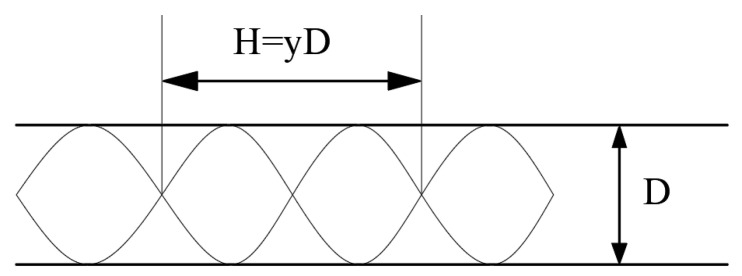
Twisted rate schematic diagram of twisted band.

**Figure 4 entropy-23-00489-f004:**
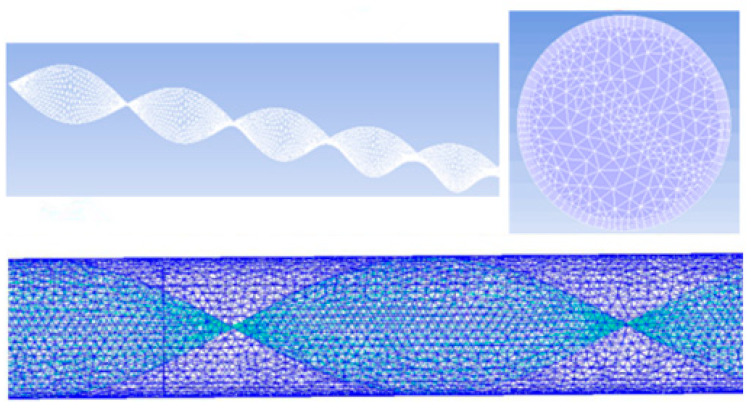
Calculation grid for long twisted band pipe section.

**Figure 5 entropy-23-00489-f005:**
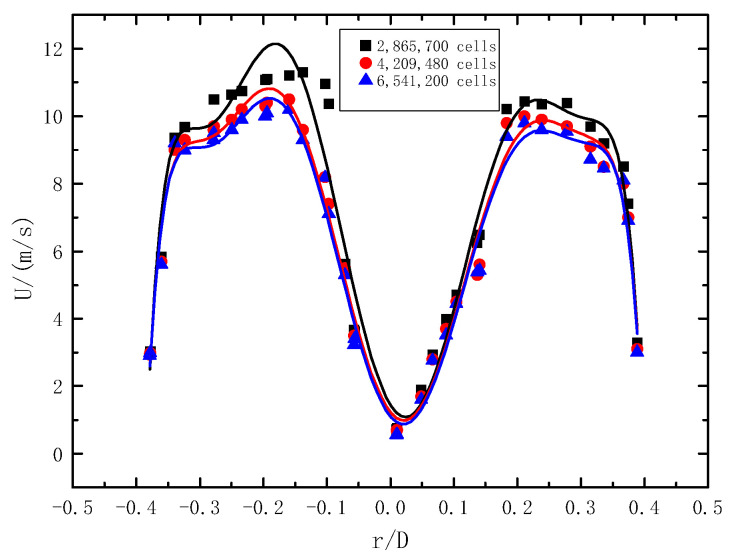
The velocity distribution on pipeline section at Z = 5D with different mesh sizes.

**Figure 6 entropy-23-00489-f006:**
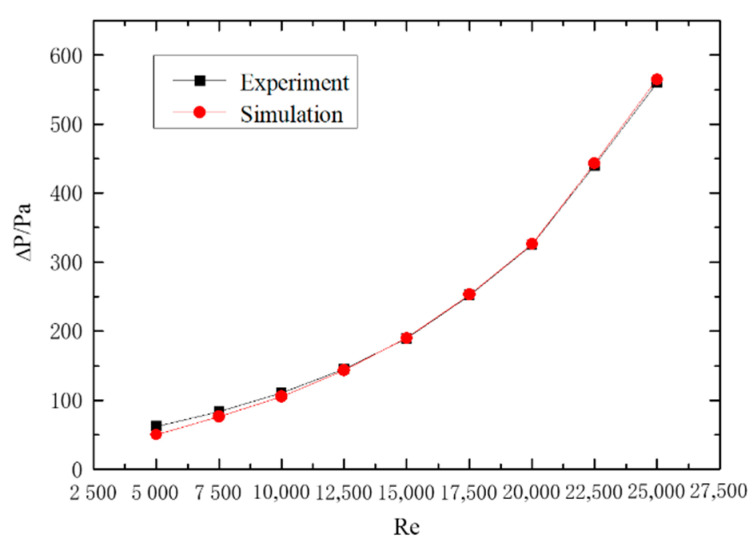
The effect of Reynolds number on pressure drop.

**Figure 7 entropy-23-00489-f007:**
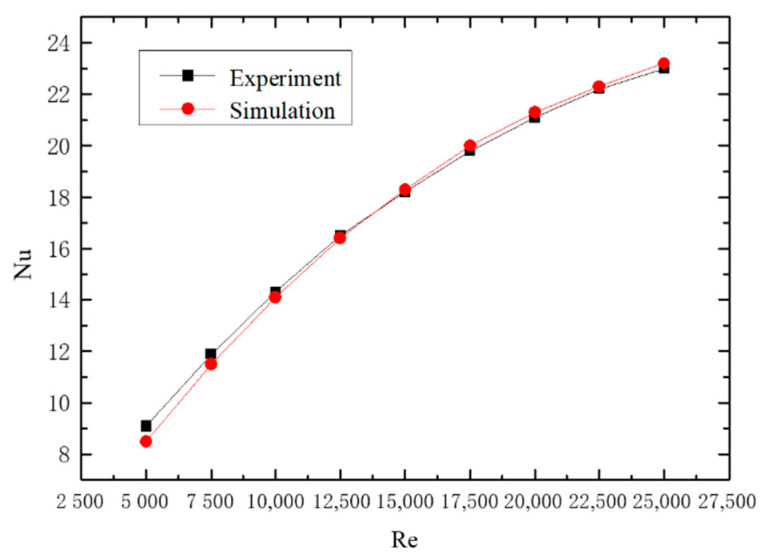
The effect of Reynolds number on Nusselt number.

**Figure 8 entropy-23-00489-f008:**
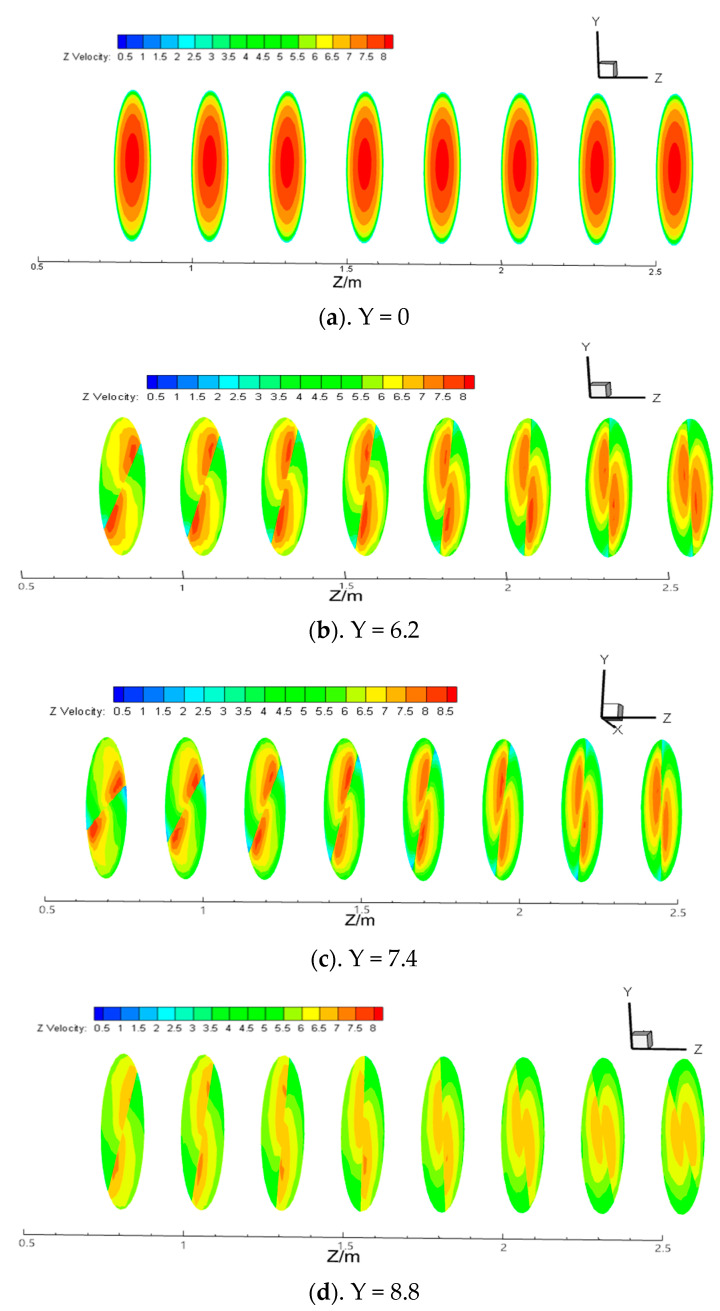
Velocity distribution clouds at different sections under Re = 15,000 and different twisted rates.

**Figure 9 entropy-23-00489-f009:**
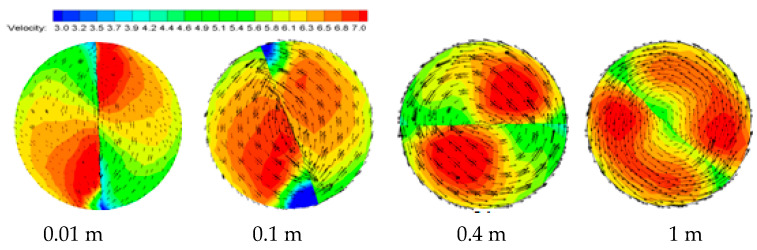
Velocity and vector distribution nephogram in the long twisted band Y = 6.2.

**Figure 10 entropy-23-00489-f010:**
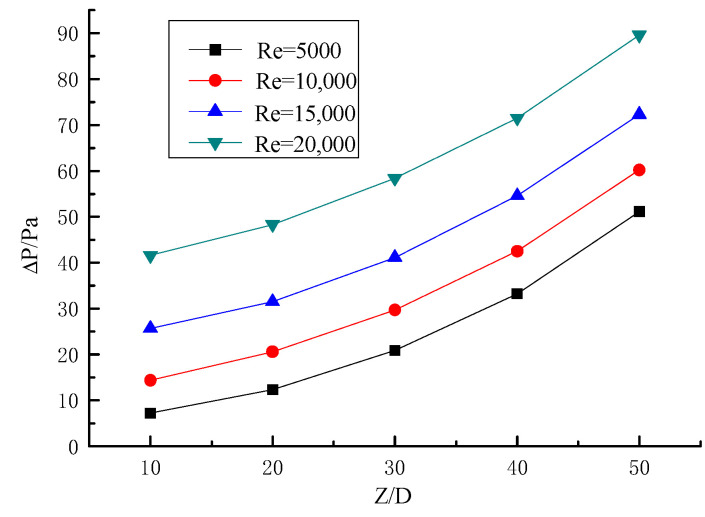
Relationship between Reynolds number and pressure drop.

**Figure 11 entropy-23-00489-f011:**
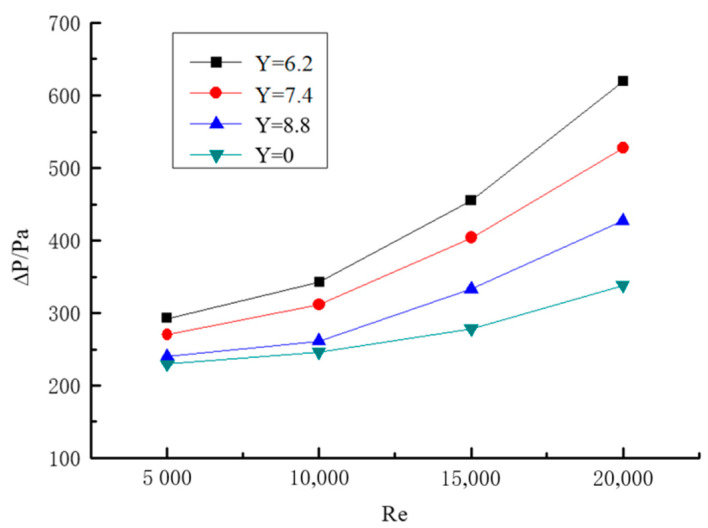
Relationship between twisted rate and pressure drop.

**Figure 12 entropy-23-00489-f012:**
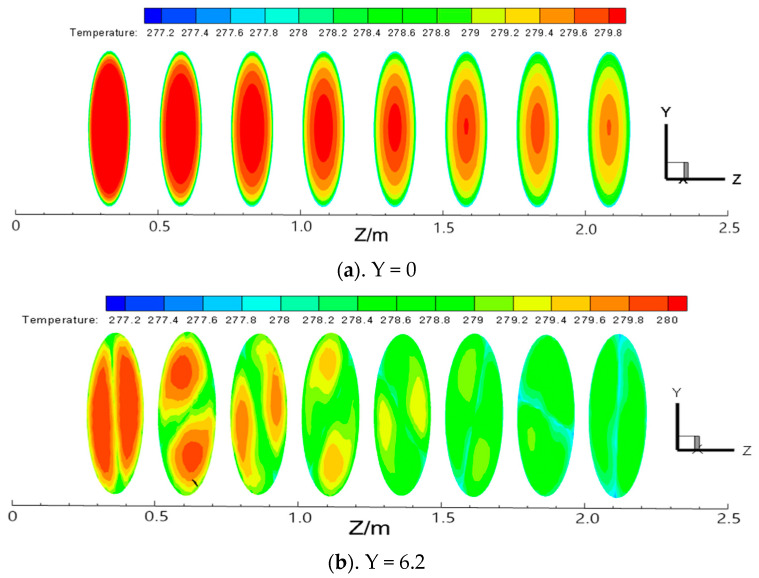
Temperature nephogram of each section in long twisted band pipeline with different twisted rate at Re = 20,000.

**Figure 13 entropy-23-00489-f013:**
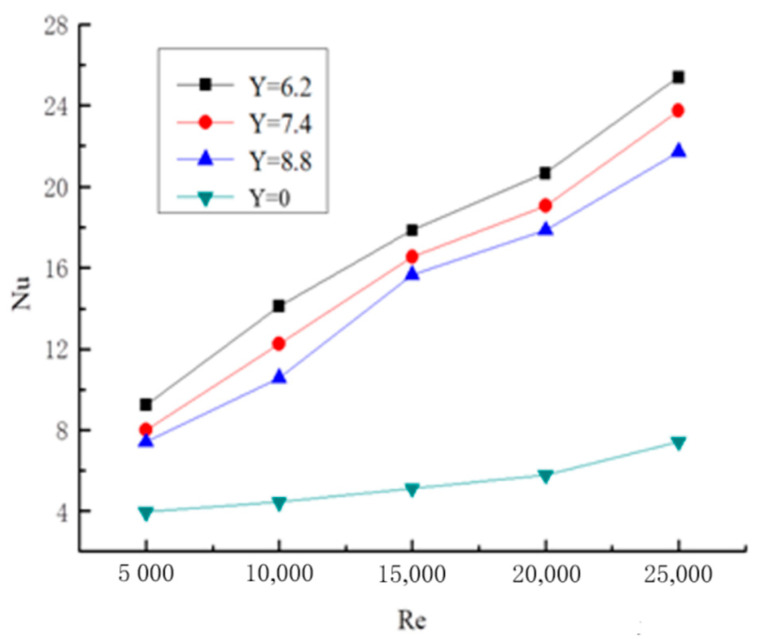
Variation curve of Nusselt number with Re in a long twisted band pipeline.

**Figure 14 entropy-23-00489-f014:**
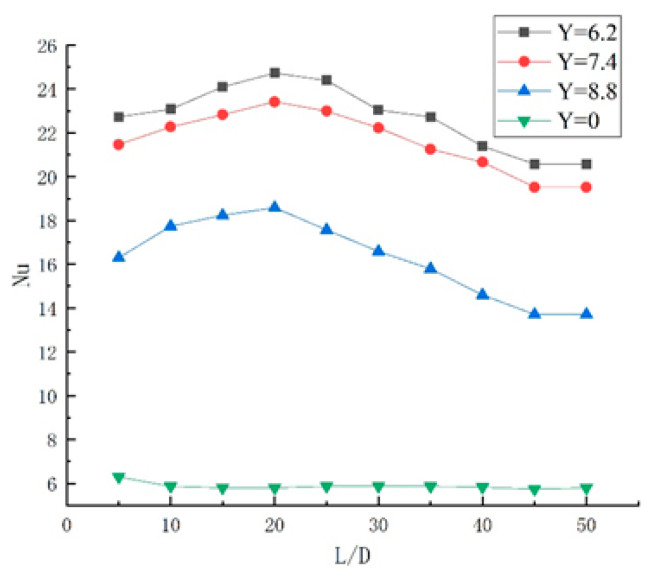
Nusselt number distribution curves of different sections under different twisted rates at Re = 20,000.

**Figure 15 entropy-23-00489-f015:**
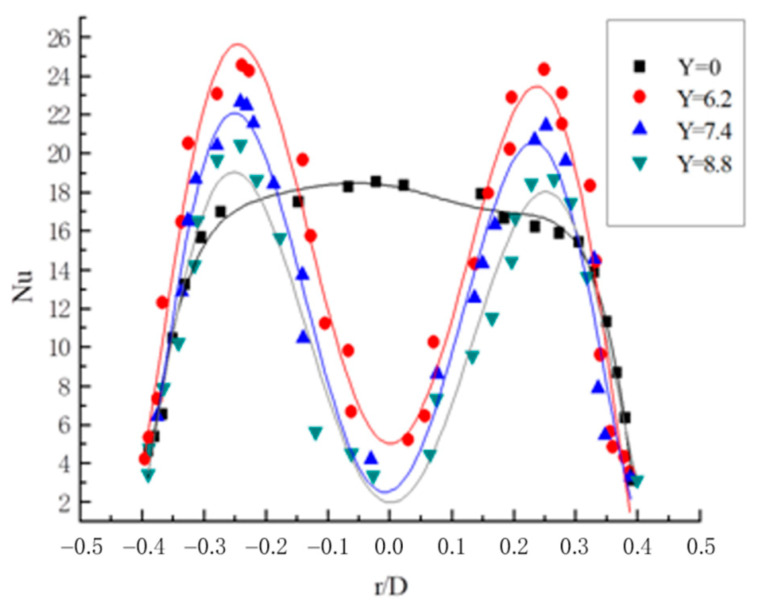
Nusselt number distribution curve on Z = 5D section at Re = 20,000.

**Figure 16 entropy-23-00489-f016:**
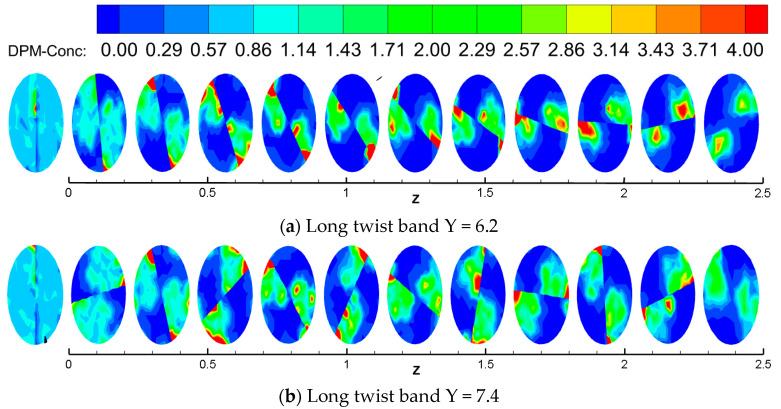
Distribution of particle concentration at different sections of the pipeline.

**Figure 17 entropy-23-00489-f017:**
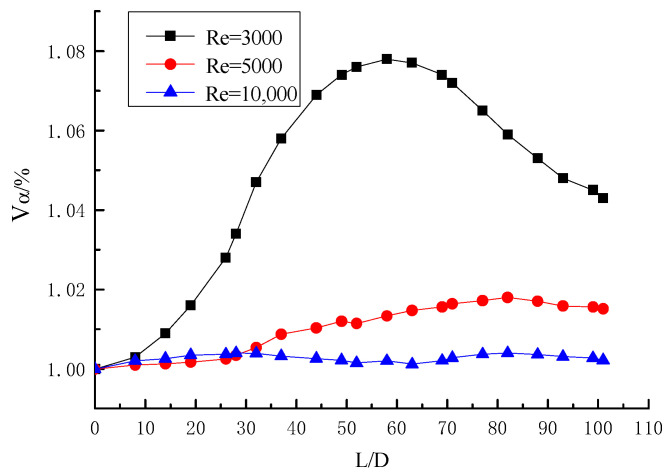
Volume fraction distribution curve of particles along pipeline at Y = 6.2.

**Figure 18 entropy-23-00489-f018:**
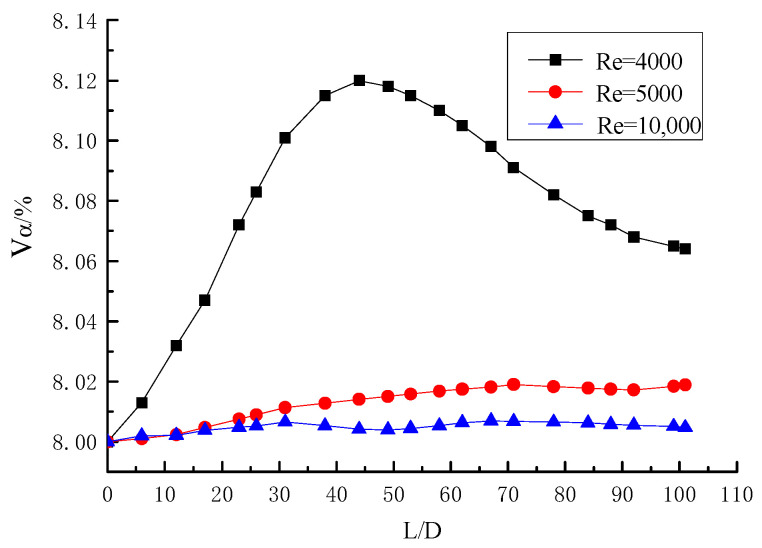
Volume fraction distribution curve of particles along pipeline at Y = 6.2, α_0_ = 8%.

**Figure 19 entropy-23-00489-f019:**
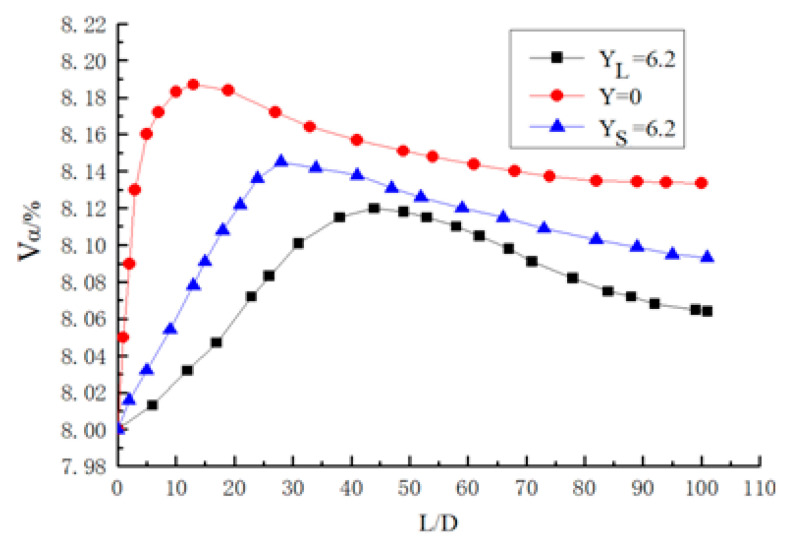
Volume fraction distribution curve of different twisted band particles along pipeline at Re = 4000, α_0_ = 8%.

**Table 1 entropy-23-00489-t001:** Table of experimental parameters for numerical simulation.

Particle Concentration (%)	Particle Diameter (m)	Twist with Twist Rate Y	Reynolds Number Re
1	0.01	0	5000
2	0.02	6.2	10,000
4	0.03	7.4	15,000
6	0.06	8.8	20,000
8	0.06	8.8	20,000

## Data Availability

The data used to support the findings of this study are available from the corresponding author upon request.

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
