# Peer review of "Numerical Simulation Study on Flow Laws and Heat Transfer of Gas Hydrate in the Spiral Flow Pipeline with Long Twisted Band"

_entropy, 2021, doi:10.3390/e23040489_

Round 1

Reviewer 1 Report

Dear authors,
The publication is interesting, but still needs to be improved. For example:
We have Figure 1 and Figure 3 is missing from Figure 2.
The velocity inlet for four different reynolds numbers is 5000, 10000, 15000 and 20000 respectively. - I do not understand this sentence. Reynolds write with a capital letter..
The initial concentration of hydrate was given as 1%, 2%, 4%, 6% and 8% ...  this apply to mass or volume concentration?
Why in table 1 there is a concentration from 1 to 6%? Where is 8?
Why is particle size related to concentration?
Figure 11 cannot be read. The quality is terrible.

Author Response

Point 1: We have Figure 1 and Figure 3 is missing from Figure 2.

Response 1: In line 151, Figure 2 has been added.

Point 2: The velocity inlet for four different reynolds numbers is 5000, 10000, 15000 and 20000 respectively. - I do not understand this sentence. Reynolds write with a capital letter..

Response 2:  This sentence is not clear, and I modified the sentence. The Reynolds Numbers is 5000, 10000, 15000 and 20000 respectively. The modified sentence is in line 157.

Point 3: The initial concentration of hydrate was given as 1%, 2%, 4%, 6% and 8% ... this apply to mass or volume concentration?

Response 3: The initial concentration of hydrate is the volume concentration. The modified sentence is in line 172.

Point 4:  Why in table 1 there is a concentration from 1 to 6%? Where is 8?

Response 4: There is a concentration from 1 to 8% in Table 1. I have modified the table. It is in line 188.

Point 5: Why is particle size related to concentration?

Response 5: The particle size is 0.02mm, and it is not related to concentration.

Point 6: Figure 11 cannot be read. The quality is terrible.

Response 6: I use a high quality picture. The modified content is in line 381.

Reviewer 2 Report

Dear Authors,

Many thanks for your submitted work. I found it really interesting. However, some changes are mandatory prior to any further actions. Please find below comments on your work:

  • Please before any action, revise your paper completely to make all sentences connected and remove grammar mistakes and typos. As a sample: look at your conclusions number 4. It is not easy to read! What was your point? Similar to this one there are many not connected sentences could be found in your manuscript.  
  • Lets go to Abstract, In Abstract first of all you need to say a brief introduction in one or 2 sentences that what is the application of such research, then a methodology and how you solve them and then you go to your results. Not all results! just some important ones that you can attract the researchers attention.
  • All abbreviations need proper explanations. Please revise them accordingly.
  • In introduction, I believe your introduction is very good and reasonable. The only issue is the usage of up to date references are very few. I have checked the reference list only one 2020 paper is included which I believe you could find more recent papers! My suggestion is to include these references which are really close to your topic: 1- "Hamed Arjmandi, Pezhman Amiri, Mohsen Saffari Pour,
    Geometric optimization of a double pipe heat exchanger with combined vortex generator and twisted tape: A CFD and response surface methodology (RSM) study,
    Thermal Science and Engineering Progress,
    Volume 18, 2020, https://doi.org/10.1016/j.tsep.2020.100514. 2- " Poornodaya Venkata Krishna Varma Kola, Srinivas Kishore Pisipaty, Siva Subrahmanyam Mendu, Rajesh Ghosh,
    Optimization of performance parameters of a double pipe heat exchanger with cut twisted tapes using CFD and RSM,
    Chemical Engineering and Processing - Process Intensification,
    Volume 163, 2021, https://doi.org/10.1016/j.cep.2021.108362. 
  • Please mention your CFD software and also add a proper reference to your reference lists! I couldnt see with which software you did simulations!
  • Please add the dimensions of Temperature and Velocity near the colorful legends for your contours. 

Author Response

Point 1: Please before any action, revise your paper completely to make all sentences connected and remove grammar mistakes and typos. As a sample: look at your conclusions number 4. It is not easy to read! What was your point? Similar to this one there are many not connected sentences could be found in your manuscript.

Response 1: I make sentences connected and remove grammar mistakes and typos. The modified contents are in line 10, 14, 15, 19, 24, 29, 35, 42, 44, 45, 51, 53, 60, 62, 64, 65, 67, 68, 75, 83, 94, 95, 108, 123, 125, 138, 160, 161, 167, 191, 192, 193, 199, 200, 202, 240, 267, 269, 273, 283, 299, 302, 316, 322, 326, 328, 329, 330, 332, 360, 361, 371, 373, 376, 384, 388, 395, 409, 410, 429, 439, 446, 451, 454, 459, 483, 497, 502, 510, 550, 551, 552, 553 and 554.

Point 2: Lets go to Abstract, In Abstract first of all you need to say a brief introduction in one or 2 sentences that what is the application of such research, then a methodology and how you solve them and then you go to your results. Not all results! just some important ones that you can attract the researchers attention.

Response 2: I have added some introduction on application of my research, a methodology and how I solve them. The modified contents are in line 10.

Point 3: All abbreviations need proper explanations. Please revise them accordingly.

Response 3: I have added proper explanations on all abbreviations, and the modified contents are in line 60, 61, 95, 124, 138 and 199.

Point 4: In introduction, I believe your introduction is very good and reasonable. The only issue is the usage of up to date references are very few. I have checked the reference list only one 2020 paper is included which I believe you could find more recent papers! My suggestion is to include these references which are really close to your topic: 1- "Hamed Arjmandi, Pezhman Amiri, Mohsen Saffari Pour, Geometric optimization of a double pipe heat exchanger with combined vortex generator and twisted tape: A CFD and response surface methodology (RSM) study, Thermal Science and Engineering Progress, Volume 18, 2020, https://doi.org/10.1016/j.tsep.2020.100514. 2- " Poornodaya Venkata Krishna Varma Kola, Srinivas Kishore Pisipaty, Siva Subrahmanyam Mendu, Rajesh Ghosh, Optimization of performance parameters of a double pipe heat exchanger with cut twisted tapes using CFD and RSM, Chemical Engineering and Processing - Process Intensification, Volume 163, 2021, https://doi.org/10.1016/j.cep.2021.108362.

Response 4: Dear reviewers and editors, thank you very much for your valuable suggestions and comments. I have added some the latest references. The modified contents are in line 118 and 623.

Point 5: Please mention your CFD software and also add a proper reference to your reference lists! I couldnt see with which software you did simulations!

Response 5: Dear reviewers and editors, thank you very much for your valuable suggestions and comments. I have added some introductions on CFD software. The modified contents are in line 129.

Round 2

Reviewer 1 Report

Dear authors, it can be seen that you have made many positive changes to the manuscript. In my opinion, it is suitable for publication.

Reviewer 2 Report

Dear Authors,

Many thanks for your efforts to address all of my comments.

This manuscript is a resubmission of an earlier submission. The following is a list of the peer review reports and author responses from that submission.

Round 1

Reviewer 1 Report

The work is interesting. Although it deals with topics already well known and described in the literature. An example is a ready model, e.g. in the comsol multiphysics application.
In static mixers, also called motionless or in-line mixers, a fluid is pumped through a pipe containing stationary blades. This mixing technique is particularly well suited for laminar flow mixing because it generates only small pressure losses in this flow regime.

Therefore, I believe that the introduction should be developed further. To highlight why this work is different from the rest.

Reviewer 2 Report

The authors presented their numerical research on a spiral flow and its particle and heat transport in pipe with twisted band. They seemed to carry out experimental counterparts to verify their numerical results. Their simulation was executed using a commercial software and run in a framework of RANS simulation with the standard k-ε model. Although a study on such complicated pipe-flow systems is practically important and valuable, I feel that this work provides little knowledge to this academic/industrial field and lacks a scientific novelty. In particular, the current paper is just a case report, which is not suitable for academic journals. In addition, regarding the methodology and its description, the present paper does not reach the journal quality. Therefore, I cannot recommend its publication in the journal of Entropy.

More specific comments are follows:

  • It is well-known that the standard k-ε model is valid for high Reynolds number and non-swirling flow. The Reynolds number of the present system is rather low and there exists a swirling motion due to twisted band. Therefore, one should check carefully the dependence on turbulence models.
  • The quality of English is very poor. A skillful native proofreading is required.
  • The introduction does not provide sufficient background and relevant references. The authors should survey literature more world wise: All the cited papers are from Chinese groups.
  • There still remain several texts of the template file. The authors should check the manuscript carefully before submission.
  • The problem setting is very unclear. For instance, in page 5, the authors say “the gravity direction is along the -y axis”, but the y-direction is not defined. What is the meaning of “the gravitational acceleration in the z direction” in line 255? Equation (3) is incorrect. The description of the used model is insufficient: what are G_k, Y_k, S_k, and S_ε in Eqs. (5) and (6)? What is and how much is d_t in Eq. (8)? In Section 2.4, the authors mention about RNG k-ε turbulence model: why? What are ε_p, ε_m, ε_k, and ε_ε?
  • Line 282-286 and line 286-290 are completely same.
  • More detailed explanation for the experimental setup and measurement method are required in Section 2.5.2.
  • Why did the authors define the Reynolds number based on the “particle” velocity, “pipe diameter”, and the “gas” viscosity? Such a definition is meaningless.
  • The definition of Nusselt number is missing.
  • The difference between YL and YS is unclear.

Reviewer 3 Report

Review on the manuscript entropy-1137783

entitled

“Numerical Simulation of Spiral Flow and Heat Transfer in Hydrate

Pipeline with Long Twisted Band”

The work presents numerical investigation on flow of hydrates in the spiral flow. The flow conditions are obtained using long twisted band insert. The numerical model used is discrete phase model to take into account flowing particles. The topic is within the range of interest of the Entropy however it does not adhere to the journal’s standards. Some issues have to be improved as well:

  • The manuscript seems to be poorly prepared (e.g. lines from 144) and it should not be taken under review.
  • Figures are not high quality
  • Grammar faults are present in the text – consider native speaker check (e.g. sentence started from line 129).
  • 1 and 1 (!) are not readable and vague.
  • Boundary conditions are not well described.
  • Meshes are rather poor and not explained.
  • The details of Nu calculations are not provided.